# Analysis of Serum Markers of Perioperative Brain Injury and Inflammation Associated with Endovascular Treatment of Intracranial Aneurysms: A Preliminary Study

**DOI:** 10.3390/brainsci13091308

**Published:** 2023-09-11

**Authors:** Mikołaj Zimny, Piotr Paździora, Damian Kocur, Bartłomiej Błaszczyk, Daria Gendosz de Carrillo, Jan Baron, Halina Jędrzejowska-Szypułka, Adam Rudnik

**Affiliations:** 1Department of Neurosurgery, Medical University of Silesia in Katowice, 40-055 Katowice, Poland; 2Department of Physiology, Medical University of Silesia in Katowice, 40-055 Katowice, Poland; 3Department of Histology and Cell Pathology, Faculty of Medical Sciences in Zabrze, Medical University of Silesia in Katowice, 40-055 Katowice, Poland; 4Department of Radiodiagnostics, Interventional Radiology and Nuclear Medicine, Medical University of Silesia in Katowice, 40-055 Katowice, Poland

**Keywords:** intracranial aneurysms, endovascular embolization, brain injury, inflammation, serum markers

## Abstract

Embolization is the preferred method for treating intracranial aneurysms due to its less invasive nature. However, recent findings suggest that even uncomplicated embolization may cause structural damage to the brain through ischemic or inflammatory mechanisms. This study aimed to find possible biomarkers of brain injury and inflammation in patients suffering from intracranial aneurysms who underwent endovascular treatment by measuring serological markers indicating brain damage. The study involved 26 patients who underwent uncomplicated intravascular stenting for unruptured intracranial aneurysms between January 2020 and December 2021. Blood samples were collected before the procedure, at 6–12 h, and at 24 h after the procedure. The following protein biomarkers levels were tested with ELISA: S100B, hNSE, TNF, hsCRP, FABP7, NFL, and GP39. Statistical analysis of the results revealed significant increases in serum levels for the four biomarkers: FABP7—before 0.25 (ng/mL) vs. 6–12 h 0.26 (*p* = 0.012) and vs. 24 h 0.27 (*p* < 0.001); GP39—before 0.03 (pg/mL) vs. 6–12 h 0.64 (*p* = 0.011) and vs. 24 h 0.57 (*p* = 0.001); hsCRP—before 1.65 (μg/mL) vs. 24 h 4.17 (*p* = 0.037); NFL—before 0.01 (pg/mL) vs. 6–12 h 3.99 (*p* = 0.004) and vs. 24 h 1.86 (*p* = 0.033). These biomarkers are recognized as potential indicators of neurovascular damage and should be monitored in clinical settings. Consequently, serum levels of NFL, GP39, hsCRP, and FABP7 measured before and 24 h after endovascular procedures can serve as important markers for assessing brain damage and indicate avenues for further research on biomarkers of neurovascular injury.

## 1. Introduction

Depending on aneurysm location and morphological features, endovascular treatment (embolization) is currently the preferred treatment for patients with intracranial aneurysms due to its less invasive nature compared to the traditional neurosurgical method of clipping. Although the endovascular technique is associated with rare complications, the perioperative course for patients treated with embolization is typically favorable. However, scientific studies indicate that, even in uncomplicated cases, the incidence of microembolic lesions, also known as “silent brain injury” or SBI, in MR diffusion-weighted imaging (DWI) after endovascular coiling of intracranial aneurysms is not low. Most patients who show SBI are asymptomatic, and the clinical significance of such findings remains unclear [1,2]. Nonetheless, these observations suggest that even uncomplicated embolization of a brain aneurysm may be associated with structural damage to the brain caused by ischemic or inflammatory mechanisms. While studies in interventional radiology refer to endarterectomy procedures or stenting of the external carotid arteries, there are currently no literature data available on the determination of serological biomarkers in terms of brain damage after embolization of intracranial aneurysms for vascular neurosurgery procedures [3,4,5]. Therefore, more research is needed to explore this area. Based on scientific research in vascular surgery, the following potential serological biomarkers of brain injury and inflammation have been identified: S100B protein, human neuron-specific enolase (hNSE), tumor necrosis factor (TNF), high-sensitivity C-reactive protein (hsCRP), FABP7 protein, neurofilament protein L (NFL), and human cartilage glycoprotein 39 (GP39).

The objective of this study was to find possible biomarkers of brain injury and inflammation in patients suffering from intracranial aneurysms who underwent endovascular treatment. Specifically, we measured serological biomarkers indicating damage to the nervous tissue or its inflammation previously reported in the literature in patients with unruptured cerebral aneurysms who underwent uncomplicated stent-only endovascular treatment. These results serve as a preliminary step towards further research into brain damage and inflammation markers in individuals after neuroendovascular procedures—diagnostic 3D-DSA and aneurysm embolization.

## 2. Materials and Methods

This study included patients hospitalized in the Neurosurgery Department at the University Clinical Center of Silesia in Katowice between 2020 and 2021 due to unruptured intracranial aneurysm.

The study was conducted in accordance with the Declaration of Helsinki, approved by the Ethics Committee of Medical University of Silesia in Katowice (PCN/0022/KB1/138/I/19/20), and all participating subjects provided written informed consent.

### 2.1. Study Population

The study involved 26 patients who received intravascular stenting for an unruptured intracranial aneurysm. The treatment was recommended by an experienced interventional radiologist based on the prior three-dimensional digital subtraction angiography (3D-DSA). As per the treatment protocol, every patient underwent follow-up imaging 6 and 12 months after the procedure. None of the patients showed signs of recanalization or the need for re-embolization or additional treatment during the follow-up period. Patients who experienced any complications during the procedure, such as neurological deficit, stroke, or abnormal wound healing, were excluded from the study. Additional exclusion criteria were age under 18 years or over 75 years, presence of multiple cerebral aneurysms or any other malformations affecting cerebral blood flow, severe clinical condition, pregnancy, and family history of intracranial aneurysms or genetical disorders associated with an increased risk of cerebral aneurysm formation (e.g., autosomal dominant polycystic kidney disease or neurofibromatosis type I).

The majority of the study participants were women (88.5%; 23/26) with a mean age of 57 years (min: 27; max: 73). More than half of the patients had hypertension (61.5%; 16/26), and half of them (50%; 13/26) reported mild neurological symptoms upon admission (headache: 9; vertigo: 4). Most patients did not show any neurological deficits upon admission (84.6%; 22/26). No complications were observed after the procedure, and all patients had a good general condition during the postoperative period. None of them showed any signs of ischemic or hemorrhagic stroke.

### 2.2. Sampling of Serum and Biomarker Analysis

Blood samples were collected three times. The first blood sample was taken before the embolization procedure, the second between 6 and 12 h after embolization, and the third at 24 h after the embolization. Approximately 5 mL of blood was taken each time, and blood samples were incubated 30–45 min to allow clotting. the samples were then centrifuged at 3000 rpm for 15 min at room temperature. The supernatant was collected and pipetted into aliquots (500 µL). Samples were stored at −80 °C until further analysis.

The protein level of S100B, hNSE, TNF, hsCRP, FABP7, NFL, and GP39 were tested in 78 samples in total using a commercially available enzyme-linked immunosorbent assay (ELISA) kit for S100B/hNSE/TNF/hsCRP/FABP7/NFL and GP39 (Tecomedical Osteogroup/Quidel, San Diego, CA, USA).

### 2.3. Statistical Analysis

The obtained results were statistically analyzed using the Statistica 13.3 program (StatSoft, Tulsa, OK, USA). The normal distribution of the analyzed variables was assessed using the Shapiro–Wilk test. Since all distributions significantly deviated from normal distribution, Friedman and Wilcoxon tests for paired samples were used for further analysis to assess the significance of changes in parameter levels over time. *p*-values < 0.05 were considered statistically significant. A minimum sample size to achieve a significance power of 80% and alpha value of 0.05 was calculated to be between 18 and 26 depending on the analyzed variable. The results were presented graphically using box plots, with quartiles (including the median) and minimum and maximum values indicated.

## 3. Results

### 3.1. S100B

There were no significant differences in serum S100B (pg/mL) levels between all three measurements: median 40.82 (interquartile range 23.641–60.916), median 37.43 (interquartile range 30.023–60.205), and median 49.59 (interquartile range 25.336–68.645) (*p* = 0.840, *p* = 0.882, and *p* = 0.840, respectively). The analysis of variance also showed that there were no significant differences in serum S100b levels between all three recorded measurements (*p* = 0.961). The serum S100B levels are presented in Figure 1.

### 3.2. hNSE

There were no significant differences in serum hNSE (ng/mL) levels between all three measurements: before median 14.63 (interquartile range 12.921–17.278), 6–12 h median 13.19 (interquartile range 10.585–15.002), and 24 h median 11.97 (interquartile range 10.889–17.653) (*p* = 0.367, *p* = 0.925, and *p* = 0.397, respectively). The analysis of variance also showed that there were no significant differences in serum hNSE levels between all three recorded measurements (*p* = 0.619). The serum hNSE levels are presented in Figure 2.

### 3.3. TNF

There were no significant differences in serum TNF (pg/mL) levels between all three measurements as results were close to 0 (*p* = 0.068, *p* = 1.0, and *p* = 0.501, respectively). The analysis of variance also showed that there were no significant differences in serum TNF levels between all three recorded measurements (*p* = 0.691). The serum TNF levels are presented in Figure 3.

### 3.4. hsCRP

hsCRP serum levels (μg/mL) were significantly higher 24 h after the procedure compared to levels before treatment: median 4.17 (interquartile range 1.765–8.158) vs. median 1.65 (interquartile range 0.931–3.401) (*p* = 0.037), respectively. There were no statistically significant differences between I vs. II and II vs. III records. The analysis of variance also indicated that there were significant differences in serum hsCRP levels between all three recorded measurements (*p* = 0.022). The serum hsCRP levels are presented in Figure 4.

### 3.5. FABP7

FABP7 serum levels (ng/mL) were significantly higher after the procedure compared to levels before treatment: before median 0.25 (interquartile range 0.245–0.261) vs. 6–12 h median 0.26 (interquartile range 0.256–0.291) (*p* = 0.012) and vs. 24 h median 0.27 (interquartile range 0.263–0.309) (*p* < 0.001). The analysis of variance also indicated that there were significant differences in serum FABP7 levels between all three recorded measurements (*p* < 0.001). The serum FABP7 levels are presented in Figure 5.

### 3.6. NFL

NFL serum levels (pg/mL) were significantly higher after the procedure compared to levels before treatment: before median 0.01 (interquartile range 0.001–0.305) vs. 6–12 h median 3.99 (interquartile range 0.156–33.499) (*p* = 0.004) and vs. 24 h median 1.86 (interquartile range 0.009–20.015) (*p* = 0.033). The analysis of variance also indicated that there were significant differences in serum NFL levels between all three recorded measurements (*p* = 0.008). The serum NFL levels are presented in Figure 6.

### 3.7. GP39

The study showed that GP39 serum levels (pg/mL) were significantly higher after the procedure compared to levels before treatment: before median 0.03 (interquartile range 0.004–0.081) vs. 6–12 h median 0.64 (interquartile range 0.011–8.456) (*p* = 0.011) and vs. 24 h median 0.57 (interquartile range 0.098–4.136) (*p* = 0.001). The analysis of variance also indicated that there were significant differences in serum GP39 levels between all three recorded measurements (*p* = 0.003). The serum GP39 levels are presented in Figure 7.

## 4. Discussion

S100B is a structural peptide of the S100 protein family, which belongs to a group of calcium-binding proteins primarily expressed in the central nervous system. S100B has level-dependent effects on neuronal tissue; at lower concentrations, it stimulates neurite outgrowth and enhances neuronal survival during development, while higher levels stimulate the expression of proinflammatory cytokines. Increased concentrations of S100B have been observed in various clinical conditions such as brain trauma [6,7] and ischemia [8,9], which may be due to the destruction of astrocytes. A single serum level of S100B can identify and stratify the severity of traumatic brain injury (TBI) [10]. In neurodegenerative, inflammatory, and psychiatric diseases, increased levels of S100B may be caused by secreted S100B or released from damaged astrocytes [11]. Brightwell et al. [12] observed a significant elevation in S100B levels among patients who developed postoperative neurological deficits after undergoing carotid artery stenting (CAS), implicating increased perioperative microembolization as the likely cause. In contrast, our study failed to detect any significant differences in serum S100B levels between patients with and without postoperative neurological deficits, which suggests that microembolization was not a contributing factor in our cohort.

Human neuron-specific enolase (hNSE) is an isoenzyme of glycolytic enolase and a major protein in the brain, which makes it a promising marker for neuronal injury. However, hNSE is also expressed in neuroendocrine tissue, erythrocytes, and platelets [13]. Cheng et al. [14] conducted a meta-analysis on the prognostic value of serum hNSE in patients with TBI and found that higher hNSE concentrations were significantly associated with mortality and unfavorable outcomes. Additionally, hNSE is a moderate prognostic factor for mortality and neurological outcomes in TBI patients. Kedziora et al. [15] investigated the changes in brain-specific biomarker levels (S100B and hNSE) in patients after an aneurysmal subarachnoid hemorrhage (aSAH). Their study demonstrated a significant increase in these markers in patients following aSAH, and a direct correlation between serum levels of S100B and hNSE and the neurological outcome. Contrasting to presented studies, our results indicated that both hNSE and S100B would not work as potential biomarkers of brain damage following uncomplicated cerebral aneurysm embolization, probably due to lack of sufficient neuronal injury.

Tumor necrosis factor (TNF) is a multifunctional cytokine that plays an essential role in various cellular events such as cell survival, proliferation, differentiation, and death [16]. In the adult brain, TNF is primarily derived from glia, astrocytes, and microglia. Although its levels are low, TNF affects the central nervous system (CNS) cells in several ways, such as regulating neurotransmitter processes, playing a pivotal role in neurogenesis [17], and influencing blood–brain barrier (BBB) permeability by altering the morphology of endothelial cells in the CNS and inducing angiogenic mediators that affect vascular endothelial cell proliferation [18,19,20]. TNF has been associated with the progressive inflammatory process that promotes aneurysm formation, progression, and rupture [21,22,23]. Due to its complex and multifaceted role in the CNS, TNF may be a potential marker for stroke and brain injury [24,25]. Our study showed that there was no significant elevation in TNF serum levels following intracranial aneurysm stenting, suggesting that performed procedures did not induce TNF-mediated processes including neuroapoptosis and BBB disruption.

C-reactive protein (CRP) is an acute-phase protein and serves as a biomarker of inflammation, with plasma concentrations rapidly increasing up to 100-fold or more in response to tissue injury or inflammation. High-sensitivity CRP (hsCRP) provides a more precise measurement of normal or baseline CRP concentrations and enables a measure of chronic inflammation. However, due to high intra-individual variability, a single hsCRP test may not accurately reflect an individual patient’s basal hsCRP level [26]. Nevertheless, recent studies indicate that hsCRP may serve as a useful component of acute and longitudinal biomarker panels for diagnosing and prognosticating mild traumatic TBI. Elevated hsCRP levels were also associated with poor clinical outcomes in arterial ischemic stroke patients receiving endovascular treatment [27]. Our study showed that hsCRP serum levels were significantly higher 24 h after the embolization compared to levels prior to treatment. There were no complications nor new neurological deficits following treatment; thus, since hsCRP is a very sensitive biomarker, its elevation may be sign of a performed procedure (anesthesia, epithelial tear) without any clinical importance.

Lipids, specifically fatty acids (FAs), are the main components involved in the formation of the myelin sheath. Fatty acid-binding proteins (FABPs) are a class of 14–15 kDa proteins that prevent FA aggregation and facilitate cellular transportation of long-chain FAs, allowing them to perform their functions within different cell compartments [28]. Astrocyte-expressed FABP7 controls FA uptake and transportation, signal transduction, gene transcription, and plays a crucial role in neurogenesis involving the formation of a radial fiber scaffold in the developing brain [29]. High FABP7 is a sign of a protective response to BBB disruption, suggesting that it may serve as a brain damage marker [30]. FABPs are also involved in the ischemic injury of neurovascular units, with FABP3 and FABP5 overexpressed in ischemic neurons, mediating oxidative stress and participating in ischemic neuronal death. FABP4 expressed in microglia stimulates the expression and release of pro-inflammatory cytokines (IL-1β, IL-6, TNFα) and matrix metalloproteinase MMP-9, which activate extrinsic apoptotic signals in ischemic neurons and disrupt the BBB. Additionally, FABP7 enhances the pro-inflammatory expression of astrocytes and stimulates ischemic neuronal apoptosis through inflammation [31], although our study showed that FABP7 serum levels were significantly higher after the endovascular procedure compared to levels before treatment; all patients were asymptomatic. Elevated FABP7 and hsCRP levels may indicate damage to the neurovascular unit, resulting in BBB disruption and inflammation, but the observation period of this study was too short to observe any long-term complications resulting from the described mechanism.

Cytoplasmic intermediate filaments are an essential component of the cytoskeleton, classified into five subclasses: keratin filaments (found in epithelial cells), vimentin filaments (found in cells of mesenchymal origin), desmin (found in muscle cells), glial filaments (found in astrocytes), and neurofilaments (found in neurons). Neurofilaments, consisting of three neuron-specific proteins (NFL 68 kDa, NFM 125 kDa, and NFH 200 kDa), internexin, and peripherin, are crucial for maintaining the structural integrity of axons. Neurofilament protein L (NFL), also known as the neurofilament light chain, is encoded by the NEFL gene in humans [32,33]. Measuring NFL levels in plasma and CSF is a reliable way to monitor axonal damage or neurodegeneration, as neurofilaments are crucial in the proper radial growth of axons [34,35,36]. Cerebrospinal fluid NFL was reported to correlate with outcome after aneurysmal subarachnoid hemorrhage [37] followed by the observations of increased CSF NFL levels in acute ischemic stroke [38]. Gisslen et al. [39] suggested that most of the NFL in peripheral blood is CNS-derived and could be used as a proxy measure for CSF NFL levels. Our study found significantly higher NFL serum levels 6–12 and 24 h after the procedure compared to admission levels, which may indicate structural damage to neurofilaments of the astrocyte-building neurovascular unit resulting from endovascular treatment for intracranial aneurysms.

Human cartilage glycoprotein 39 (GP39), also known as chitinase-3-like protein 1, is a mammalian glycoprotein. GP39 is synthesized and secreted by various cells, including macrophages, neutrophils, synoviocytes, chondrocytes, fibroblast-like cells, smooth muscle cells, and tumor cells. Xu et al. [40] found that serum GP39 levels were elevated in patients with cerebrovascular diseases, and that serum levels of GP39 were significantly lower in healthy control subjects than in patients with acute ischemic or hemorrhagic stroke and transient ischemic attack. Park et al. [41] reported that serum GP39 levels were associated with infarct volume, stroke severity, and neurological outcome in acute ischemic stroke patients. Jiang et al. [42] observed that GP39 is associated with inflammation and severity of intracerebral hemorrhage (ICH) and may independently predict long-term clinical outcomes of ICH. These authors concluded that GP39 may be a useful diagnostic and prognostic biomarker for cerebrovascular disease. Our study revealed that GP39 is increased in patients after aneurysm stenting and that GP39 may be a useful biomarker for monitoring cerebrovascular disease and assessing the efficacy of endovascular treatments.

Although the mechanism of the elevations observed in our study is currently unknown, we hypothesize that damage to the neurovascular unit during endovascular treatment may be a result of mechanical irritation and/or temporary changes in the local hemodynamic environment. Damage to the neurovascular unit results in BBB disruption (FABP7), local inflammation (hsCRP, FABP7, GP39), and astrocytic injury (NFL). Disruption of the BBB allows the abovementioned markers to freely pass into the bloodstream. Moreover, neurovascular damage triggers not only local neuroinflammation induced by cell death but also systemic inflammation caused by disruption of the BBB [43,44]. Lack of increased proinflammatory markers (TNF) indicates that the described process is self-containing. Due to the uncomplicated course of treatment and lower levels of analyzed markers than in case of TBI or SAH, it can be assumed that no significant damage to the nervous tissue occurs.

As a preliminary study, this work has several limitations that should be taken into account. Firstly, the sample size was relatively small, which may limit the generalizability of the results to a larger population. Secondly, the follow-up period was short, and it is possible that the long-term effects of the treatment were not captured. Additionally, there is a paucity of previous research studies on this topic and on possible impact of the neuroendovascular procedures on brain damage/inflammation in general, which makes it challenging to compare the findings to the existing literature. No additional imaging studies, such as CT or MRI, were performed because of the uncomplicated course of treatment. Therefore, it was impossible to determine presence and extent of potential brain damage that may have occurred and whether the treatment was completely safe. It is crucial to acknowledge these limitations when interpreting the findings of this study and to consider the need for further research with larger study group, longer follow-up periods, and more comprehensive imaging studies to obtain a more comprehensive understanding of the effects of aneurysm stenting, and neuroendovascular procedures in general, on brain damage.

In summary, the findings of our study indicate that endovascular treatment of intracranial aneurysms has a significant impact on the serum levels of NFL, GP39, hsCRP, and FABP7. These biomarkers have been recognized in the literature as potential indicators of neurovascular damage, making their monitoring an essential practice in clinical settings. Variations in the analyzed markers can signify neurovascular damage caused by stenting, followed by progressive neuroinflammation and BBB disruption. Consequently, NFL, GP39, hsCRP, and FABP7 could serve as important markers of brain damage following endovascular procedures, and their serum levels should be measured before and 24 h after treatment. However, the observed changes in serum levels of these biomarkers in patients do not necessarily require any modification of the recommendations for managing patients with unruptured intracranial aneurysms. These changes, however, indicate avenues for further research on biomarkers of neurovascular injury. Whenever possible, complete obliteration of the aneurysm is recommended, and the choice of the treatment method should involve a multidisciplinary decision based on the patient’s characteristics and the aneurysm’s features. Stent-only embolization remains a safe and preferred treatment method for cerebral aneurysms in most cases.

## Figures and Tables

**Figure 1 brainsci-13-01308-f001:**
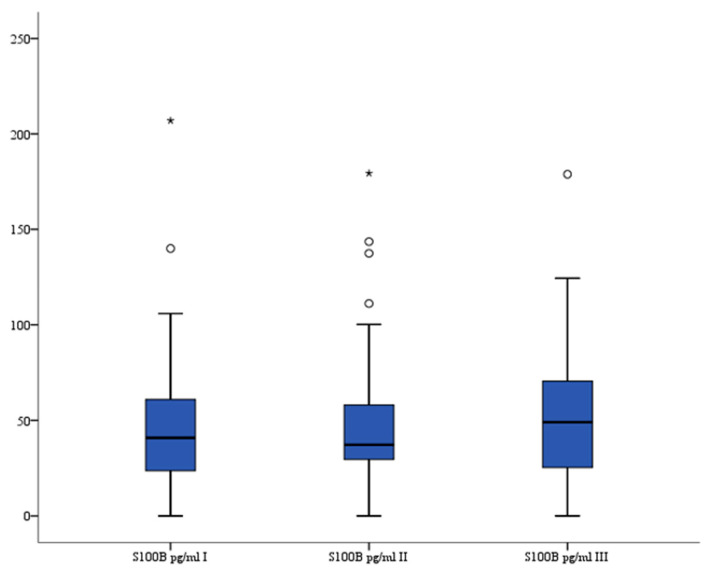
Serum S100B level in patients. °—outlier, *—far outlier.

**Figure 2 brainsci-13-01308-f002:**
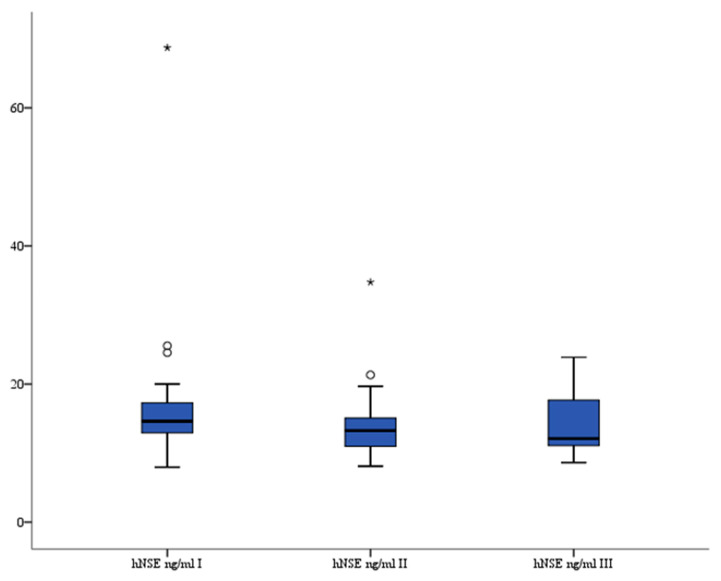
Serum hNSE level in patients. °—outlier, *—far outlier.

**Figure 3 brainsci-13-01308-f003:**
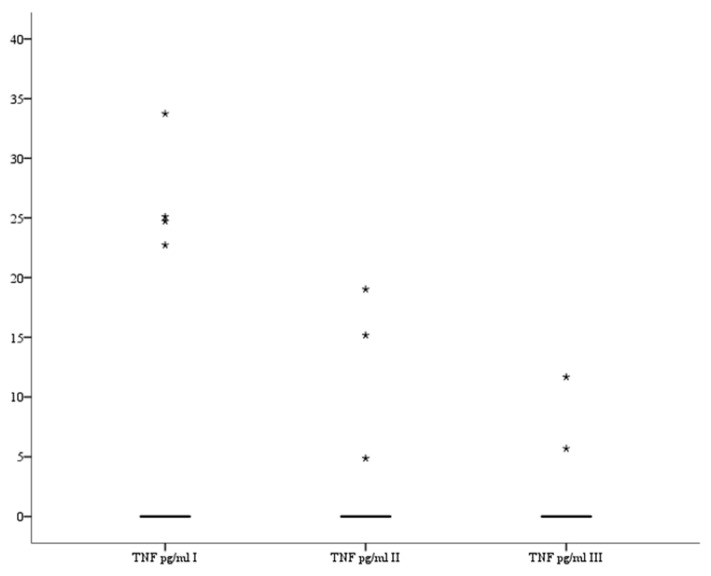
Serum TNF level in patients. *—far outlier.

**Figure 4 brainsci-13-01308-f004:**
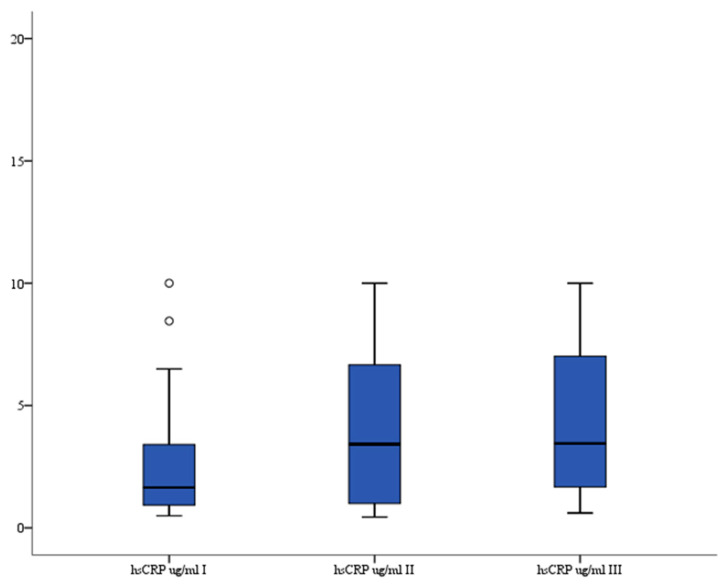
Serum hsCRP level in patients. °—outlier.

**Figure 5 brainsci-13-01308-f005:**
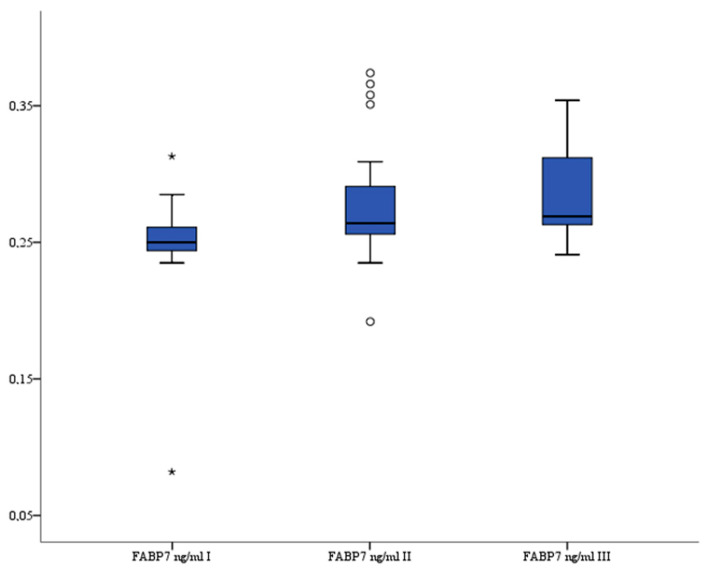
Serum FABP7 level in patients. °—outlier, *—far outlier.

**Figure 6 brainsci-13-01308-f006:**
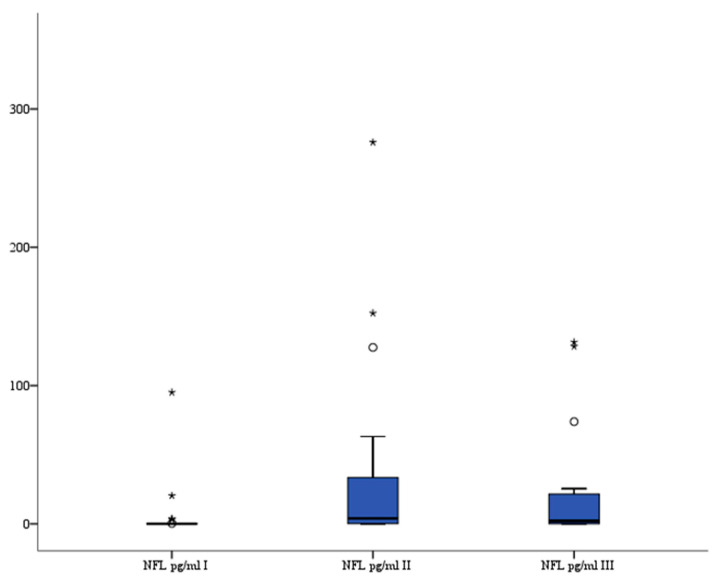
Serum NFL level in patients. °—outlier, *—far outlier.

**Figure 7 brainsci-13-01308-f007:**
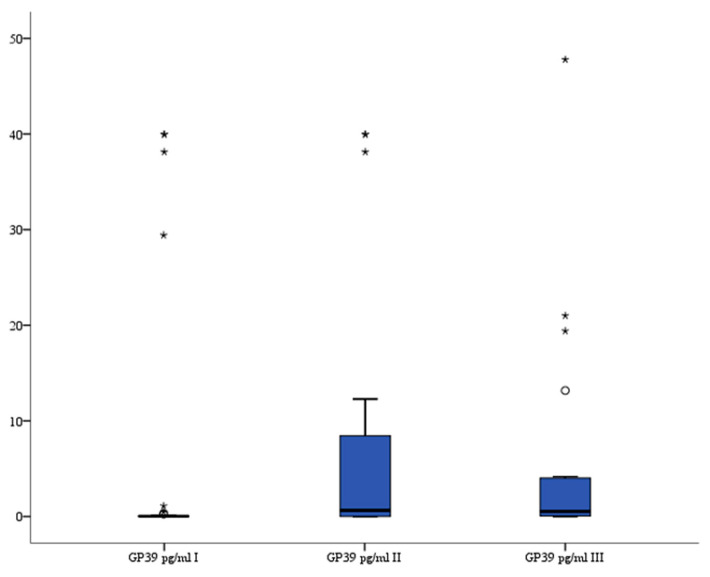
Serum GP39 level in patients. °—outlier, *—far outlier.

## Data Availability

The anonymized data presented in this study are available on request from the corresponding author. The data are not publicly available due to ethical and privacy concerns.

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
