# Peer review of "Analysis of Serum Markers of Perioperative Brain Injury and Inflammation Associated with Endovascular Treatment of Intracranial Aneurysms: A Preliminary Study"

_brainsci, 2023, doi:10.3390/brainsci13091308_

Round 1
Reviewer 1 Report
The present study about the role of inflammation serum markers associated with endovascular treatment of un-ruptured intracranial aneurysms to predict brain damage is well written and conducted; the author’s findings and final statement was in favour of testing “NFL, GP39, hsCRP, and FABP7 serum levels before and 24 hours after embolization as an important markers for assessing brain damage”.
On the other hand, few problems are encountered as follow:
1) It would be important that the authors explain the etio-pathological basis of any possible inflammation/ischemic brain damage in uncomplicated brain aneurysm embolization procedure (please explain, add details, arguments).
2) Is there any difference in impacting such inflammation serum marker level between simple flow diverter stent vs coiling embolization and etc.? (please explain, add details) ; in case of any difference could the authors explain any possible mechanisms (please discuss).
3) It would be important to know why the authors decided to test only serum markers instead also of CSF markers comparing them (please explain, add details and at this purpose consider pertinent literature as follow: Messina R, de Gennaro L, De Robertis M, Pop R, Chibbaro S, Severac F, Blagia M, Balducci MT, Bozzi MT, Signorelli F. Cerebrospinal Fluid Lactate and Glucose Levels as Predictors of Symptomatic Delayed Cerebral Ischemia in Patients with aneurysmal Subarachnoid Hemorrhage. World Neurosurg. 2023 Feb;170:e596-e602. doi: 10.1016/j.wneu.2022.11.068. Epub 2022 Nov 18. PMID: 36403937.
4) The patient cohort is quite small.
5) The results and discussion section is too long and sometimes redundant (pleaser shorten it).
Author Response
Dear Reviewer,
Thank you for taking the time to review our manuscript. We sincerely appreciate your valuable feedback and insights, which have undoubtedly contributed to improving the quality of our work. Below, we address each of your comments and suggestions:
Ad.1. Mechanisms leading to the elevation of analyzed markers have been presented and described in the reviewed Discussion section of the manuscript for each of the markers, as well as in the last paragraph serving as manuscript's Conclusions.
Ad.2. To the best of our knowledge, no studies have been conducted analyzing brain damage markers in patients undergoing coil embolization. Therefore, it is impossible to compare the analyzed markers between these groups (stent vs coil) or determine any potential mechanism.
Ad.3. The significant majority of studies on brain damage markers have so far been measured in blood/serum. Furthermore, blood collection is less invasive, carries a lower risk of complications, and can be performed by a nurse without the need for medical supervision.
Ad.4. The fact of a small sample size in the study group has been presented and described in the manuscript as one of its limitations. The presented study had a limited budget, which did not allow for a large sample size. The obtained results will enable the analysis of selected factors in a larger group of patients in the near future.
Ad.5. The Discussion section has been shortened and restructured in accordance with the suggestion. Insignificant passages and citations have been removed, improving the readability of the manuscript.
In conclusion, we are grateful for the reviewer's thorough evaluation and constructive feedback. We believe that the revisions we have made in response to your comments have significantly enhanced the quality and clarity of our manuscript. We are confident that these improvements strengthen the overall contribution of our work to the field. We hope that the revised version of the paper meets the standards of Brain Sciences and addresses all concerns effectively.
Thank you once again for your time and consideration.
Sincerely,
Mikołaj Zimny
Medical University of Silesia
Reviewer 2 Report
Major concerns:
1) Throughout the length of the manuscript, the authors refer to endovascular treatment generically by referring alternately to embolization or stenting. I would use the generic term only when endovascular treatment means both treatments, and the specific terms when only one of the two is involved
2) given that these biomarkers are nonspecific and have also been shown in traumatic brain injury, I would not want there to be a big bias in this study: that is, that these values do not depend on the stenting procedure per se, but also simply on the contrast medium used for angiography. I would like to know from the authors if there are any studies in the literature that have also separately evaluated individual biomarkers in case of stenting, coiling, or angiography only...Which would not make the study useless, but it should be specified what kind of brain damage we are evaluating
Introduction
Line 34: I would not write such a general sentence. Endovascular treatment, either stenting or coiling, may be preferred in certain districts, in elective cases, depending on the morphology of the aneurysmal sac, etc.
Materials and methods:
line 83-88: I would move this paragraph into the results section
The patients included were Stenting alone or also coiled assisted?
results:
even if not extremely relevant for the study, I would know the population characteristics (line 83-88 in methods section), and also the characteristics of the aneurysm (anterior or posterior circulation, diameter, relevant comorbidities ecc)
Discussion
the discussion section is extremely redundant and it is difficult to be oriented within it. I would focus more on the results of the study rather than describing the characteristics of individual biomarkers, which are not the subject of the study. I would also add subheadings to make it easier to read
Conclusion
There is no conclusion section
Author Response
Dear Reviewer,
Thank you for taking the time to review our manuscript. We sincerely appreciate your valuable feedback and insights, which have undoubtedly contributed to improving the quality of our work. Below, we address each of your comments and suggestions:
Comment: “Throughout the length of the manuscript, the authors refer to endovascular treatment generically by referring alternately to embolization or stenting. I would use the generic term only when endovascular treatment means both treatments, and the specific terms when only one of the two is involved”
Answer: “The manuscript has been revised according to the suggestions.”
Comment: “given that these biomarkers are nonspecific and have also been shown in traumatic brain injury, I would not want there to be a big bias in this study: that is, that these values do not depend on the stenting procedure per se, but also simply on the contrast medium used for angiography. I would like to know from the authors if there are any studies in the literature that have also separately evaluated individual biomarkers in case of stenting, coiling, or angiography only...Which would not make the study useless, but it should be specified what kind of brain damage we are evaluating”
Answer: “To the best of our knowledge, no studies have been conducted analyzing brain damage markers in patients undergoing coil embolization or 3D-DSA. Therefore, it is impossible to compare the analyzed markers between these groups (stent vs coil vs angiography) or determine any potential mechanism.”
Comment: „I would not write such a general sentence. Endovascular treatment, either stenting or coiling, may be preferred in certain districts, in elective cases, depending on the morphology of the aneurysmal sac, etc.”
Answer: “The manuscript has been revised according to the suggestions.”
Comment: “line 83-88: I would move this paragraph into the results section”
Answer: “Since the lines in question constitute a description of the studied population, we believe that they should remain in the Materials and Methods section. Additionally, presented characteristics did not impact obtained results.”
Comment: “The patients included were Stenting alone or also coiled assisted?”
Answer: “As stated in the revised text all patients underwent stent-only embolization treatment.”
Comment: “even if not extremely relevant for the study, I would know the population characteristics (line 83-88 in methods section), and also the characteristics of the aneurysm (anterior or posterior circulation, diameter, relevant comorbidities ecc)”
Answer: “Population characteristics were presented in the Materials and Methods section (Study population paragraph), the only comorbidity was hypertension present in 16 patients. 25 patients suffered from aneurysm located within the anterior (16 on internal carotid artery, 7 on anterior communicating artery and 2 on anterior cerebral artery) and 1 within posterior circulation (basilar artery). Mean neck dimension was 3,3±1,5mm (min: 1,5 and max: 8) and mean aneurysm sac volume was 70,2±72,3mm3 (min: 2,6 and max: 261,8).”
Comment: “the discussion section is extremely redundant and it is difficult to be oriented within it. I would focus more on the results of the study rather than describing the characteristics of individual biomarkers, which are not the subject of the study. I would also add subheadings to make it easier to read”
Answer: “The Discussion section has been shortened and restructured in accordance with the suggestion. Insignificant passages and citations have been removed, improving the readability of the manuscript. We believe that adding subheading may disrupt natural “flow” of the discussion.”
In conclusion, we are grateful for the reviewer's thorough evaluation and constructive feedback. We believe that the revisions we have made in response to your comments have significantly enhanced the quality and clarity of our manuscript. We are confident that these improvements strengthen the overall contribution of our work to the field. We hope that the revised version of the paper meets the standards of Brain Sciences and addresses all concerns effectively.
Thank you once again for your time and consideration.
Sincerely,
Mikołaj Zimny
Medical University of Silesia
Round 2
Reviewer 1 Report
The authors have made an effort to improve their study but unfortunately they have not answered satisfactorily to the major points raised by the reviewers.
I believe that the study has a good potential but to be improved it needs the following points to be addressed and developed:
1) The authors should state the primary and secondary if any endpoints in the methods section and explain methodologically the tools to assess them.
2) The authors once again should explain/support by etio-physio-pathological basis/evidences any possible inflammation/ischemic brain damage in uncomplicated brain aneurysm embolization procedure (please explain, add details, arguments).
3) Methodologically and statistically the authors should state the minimum number of patients to achieve a significance power of 80, 90, etc%. I understand that the budget maybe was limited but this unfortunately does not fix any scientific flaws.
Author Response
Dear Reviewer,
Thank you for taking additional time to review our manuscript for the second time. We thank the Reviewer for the thoughtful review of our work and kind words. We have thoroughly re-reviewed the manuscript and corrected any errors we came across. Below, we address each of your comments and suggestions:
Ad.1. Primary and secondary endpoints were defined as aims of the study in the Introduction paragraph: “The objective of this study was to find possible biomarkers of brain injury and inflammation in patients suffering from intracranial aneurysms who underwent endovascular treatment(…). These results serve as a preliminary step towards further research into brain damage and inflammation markers in individuals after neuroendovascular procedures – diagnostic 3D-DSA and aneurysm embolization.” with methodology described briefly in the beginning of the manuscript (“Specifically, we measured serological biomarkers indicating damage to the nervous tissue or its inflammation previously reported in the literature in patients with unruptured cerebral aneurysms who underwent uncomplicated stent-only endovascular treatment.”) and expended in the Methodology paragraph.
Ad.2. Our hypothesis regarding possible mechanism of the increase in the described biomarkers has been expanded and described in the Discussion paragraph.
Ad.3. A minimum sample size to achieve a significance power of 80% and alpha value of 0.05 was calculated to be between 18 and 26 depending on the analyzed variable. A proper text fragment was added to the manuscript.
Due to a paucity of previous research studies on this topic and on possible impact of the neuroendovascular procedures on brain damage/inflammation in general, it is challenging to compare our findings to existing literature. As stated in the manuscript, our results indicate avenues for further research on biomarkers of neurovascular injury in patients after neuroendovascular procedures – both diagnostic and treatment. Because of that we decided to change title and the general tone of the paper to the preliminary study.
Thank you once again for your time and consideration.
Sincerely,
Mikołaj Zimny
Medical University of Silesia
Reviewer 2 Report
As much as I appreciate the revisions made by the authors, major concern number 2 remains and is in my opinion insurmountable. The article claims in the title and conclusion that the stenting procedure increases concentrations of markers indicative of brain damage and therefore that this technique is a possible cause of damage. However, as already questioned in the previous version, these claims cannot be supported in the absence of control groups that can assess similar markers in patients undergoing DSA and coiling. Moreover, the authors fail to give even a plausible hypothesis in illustrating why precisely stenting should cause brain damage (moreover, assessed with nonspecific markers of damage, which could also easily be elevated simply because of the trauma caused by the catheter, or the contrast medium). In conclusion, I do not claim that the conclusions are untrue, but I do not believe that they are scientifically valid and I cannot endorse the publication of this article in this form.
Author Response
Dear Reviewer,
Thank you for taking additional time to review our manuscript for the second time. We thank the Reviewer for the thoughtful review of our work and kind words. We have thoroughly re-reviewed the manuscript and corrected any errors we came across.
Our hypothesis regarding possible mechanism of the increase in the described biomarkers has been expanded and described in the Discussion paragraph. Due to a paucity of previous research studies on this topic and on possible impact of the neuroendovascular procedures on brain damage/inflammation in general, it is challenging to compare our findings to existing literature. As stated in the manuscript, our results indicate avenues for further research on biomarkers of neurovascular injury in patients after neuroendovascular procedures – both diagnostic and treatment. Because of that we decided to change title and the general tone of the paper to the preliminary study.
We believe that our results will benefit the academic community because they introduce new facts to the ongoing discussion regarding the preferred method of treatment of unruptured cerebral aneurysms and the safety of endovascular treatment, and indicates the direction for future research in this field.
Thank you once again for your time and consideration.
Sincerely,
Mikołaj Zimny
Medical University of Silesia